

**Direct comparisons of ice cloud macro- and microphysical properties simulated by the Community Atmosphere Model version 5 with HIPPO aircraft observations**

Chenglai Wu[1,2], Xiaohong Liu[1,*], Minghui Diao[3], Kai Zhang[4], Andrew Gettelman[5], Zheng Lu[1], Joyce E. Penner[6], and Zhaohui Lin[2]

[1]*Department of Atmospheric Science, University of Wyoming, Laramie, Wyoming, USA*

[2]*International Center for Climate and Environment Sciences, Institute of Atmospheric Physics, Chinese Academy of Sciences, Beijing, China*

[3]*Department of Meteorology and Climate Science, San Jose State University, San Jose, California, USA*

[4]*Pacifit Northwest National Laboratory, Richland, Washington, USA*

[5]*National Center for Atmospheric Research, Boulder, Colorado, USA*

[6]*Department of Climate and Space Sciences and Engineering, University of Michigan, Ann Arbor, Michigan, USA*

*Corresponding to:*

Xiaohong Liu
Department of Atmospheric Science
University of Wyoming
Dept. 3038, 1000 East University Avenue
Laramie, WY 82071
Email: xliu6@uwyo.edu.



**Abstract**

In this study we evaluate cloud properties simulated by the Community

Atmosphere Model Version 5 (CAM5) using in-situ measurements from the HIAPER
Pole-to-Pole Observations (HIPPO) for the period of 2009 to 2011. The modeled
wind and temperature are nudged towards reanalysis. Model results collocated with
HIPPO flight tracks are directly compared with the observations, and model
sensitivities to the representations of ice nucleation and growth are also examined.
Generally, CAM5 is able to capture specific cloud systems in terms of vertical
configuration and horizontal extension. In total, the model reproduces 79.8% of
observed cloud occurrences inside model grid boxes, and even higher (94.3%) for ice
clouds (T≤-40°C). The missing cloud occurrences in the model are primarily ascribed
to the fact that the model cannot account for the high spatial variability of observed
relative humidity (RH). Furthermore, model RH biases are mostly attributed to the
discrepancies in water vapor, rather than temperature. At the micro-scale of ice clouds,
the model captures the observed increase of ice crystal mean sizes with temperature,
albeit with smaller sizes than the observations. The model underestimates the
observed ice number concentration ($N_i$) and ice water content (IWC) for ice crystals
larger than 75 μm in diameter. Modeled IWC and $N_i$ are more sensitive to the
threshold diameter for autoconversion of cloud ice to snow ($D_{cs}$), while simulated ice
crystal mean size is more sensitive to ice nucleation parameterizations than to $D_{cs}$.
Our results highlight the need for further improvements to the sub-grid RH variability
and ice nucleation and growth in the model.



## 1 Introduction

Cirrus clouds, the type of clouds composed of ice crystals, are one of the key
components in the climate system. Cirrus clouds cover about 30% of the globe (Wang
et al., 1996; Wylie and Menzel, 1999). They have a significant impact on the earth's
radiation balance via two different effects: scattering and reflecting the incoming
short wave solar radiation back to space, which leads to a cooling effect on the planet;
and absorbing and re-emitting terrestrial longwave radiation, leading to a warming
effect (Liou, 1986; Ramanathan and Collins, 1991; Corti et al., 2005). The net
radiative effect is thus a balance of these two effects and mainly depends on the
amount, microphysical and optical properties of cirrus clouds (Kay et al., 2006;
Fusina et al., 2007; Gettelman et al., 2012; Tan et al., 2016). Furthermore, as the
efficiency of dehydration at the tropical tropopause layer is strongly influenced by the
microphysical processes within cirrus clouds, cirrus clouds can also regulate the
humidity of air entering the stratosphere and are recognized as an important
modulator for water vapor in the upper troposphere and the lower stratosphere
(Gettelman et al., 2002; Wang and Penner, 2010; Jensen et al., 2013; Dinh et al.,

2014).

Despite their important role in the climate system, there are still large
uncertainties in the representation of cirrus clouds in global climate models (GCMs)
(Boucher et al., 2013). The uncertainties are the result of several different aspects.
First, our understanding of processes initiating the cirrus cloud formation is still
limited (DeMott et al., 2003; Kärcher and Spitchtinger, 2009; Hoose and Möhler,





2012). Ice crystals can form via the homogeneous nucleation of soluble aerosol
particles and the heterogeneous nucleation associated with insoluble or partly
insoluble aerosol particles (e.g., Hagg et al., 2003; Liu and Penner, 2005; Wang and
Liu, 2014). Homogeneous nucleation generally requires higher ice supersaturation
and occurs at temperatures colder than about -37°C. It can be fairly well represented
by nucleation theory based on laboratory results (Koop et al., 2000). Heterogeneous
nucleation is initiated by certain types of aerosols (e.g., mineral dust and biological
aerosols) that act as ice nucleating particles (INP), which can nucleate ice particles at
significantly lower ice supersaturations in the environment. Currently there are still
large unknowns about the types of aerosol, modes of action (e.g.,
immersion/condensation, deposition, contact), and the efficiencies of heterogeneous
nucleation in the atmosphere (Hoose and Möhler, 2012). Other ice microphysics (e.g.,
ice aggregation, deposition/sublimation, and sedimentation), as well as interactions
among cirrus microphysical properties, macroscopic properties (e.g., spatial extent),
and meteorological fields could further render the interpretation of observed ice cloud
properties challenging (Diao et al., 2013; Krämer et al., 2016).
In addition to our limited understanding of ice microphysical processes, it is
difficult for GCMs with coarse spatial resolution (e.g., tens to hundreds of kilometers
in the horizontal direction, and a kilometer in the vertical) to capture the sub-grid
variability of dynamical and microphysical processes that are vital for ice cloud
formation and evolution. The observed microphysical properties of cirrus clouds vary
significantly in time and space (e.g., Hoyle et al., 2005; Diao et al., 2013; Jensen et al.,



2013; Diao et al., 2014a), associated with variability in relative humidity, temperature,
and vertical wind speed. The spatial extent of clouds is represented in GCMs by
diagnosing the cloud fraction in individual model grid boxes using a parameterization.
Such a cloud fraction representation needs to be validated with observations in order
to identify model biases and to elucidate the reasons behind these biases for future
model improvement.
Two types of observational data are currently available for validating modeled
cirrus cloud properties: in-situ aircraft measurements (e.g., Krämer et al., 2009;
Lawson et al., 2011; Diao et al., 2013), and remote-sensing data from space-borne or
ground-based instruments (Mace et al., 2005; Deng et al., 2006, 2008; Li et al., 2012).
Remote-sensing data may not be directly comparable to model simulations due to the
sampling and algorithmic differences between GCM results and remote-sensing
retrievals unless a proper simulator, i.e. a so called "satellite simulator", is adopted
(Bodas-Salcedo et al., 2011; Kay et al., 2012). In-situ aircraft observations can
provide direct measurements of ice crystal properties such as ice crystal number
concentration and size distribution. In particular, these observations are a good source
of accurate and fast measurements, and thus provide a unique tool for constraining
GCM cirrus parameterizations (e.g., Zhang et al., 2013; Eidhammer et al., 2014).
However, the grid scales of GCMs are much larger than those sampled by in-situ
observations. Thus direct comparisons at model grid scales are often hindered unless
in-situ observations are adequately distributed within the grid boxes and can be scaled
up. At the micro-scale level of cirrus clouds (sub-grid scale), statistical comparisons



between model simulations and in-situ observations, especially in terms of
relationships among cloud microphysical and meteorological variables, are desirable
to provide a reliable evaluation of model microphysics (e.g., Zhang et al., 2013;
Eidhammer et al., 2014). In addition, aircraft measurements are often limited in their
spatial and temporal coverage, which in some sense limits the scope of
model-observation comparisons that can be conducted.

Previous studies have focused on the evaluation of cirrus clouds from

free-running GCM simulations against in-situ observations (e.g., Wang and Penner,
2010; Zhang et al., 2013; Eidhammer et al., 2014). However, since the model
meteorology was not constrained by conditions that were representative of the time of
the observations, the model biases could not be exclusively ascribed to errors in the
cirrus parameterizations. Recently, a nudging technique has been developed to allow
the simulated meteorology to be more representative of global reanalysis/analysis
fields, and thus the comparison between model simulations and observations is more
straightforward for the interpretation and attribution of model biases (Kooperman et
al., 2012; Zhang et al., 2014). In such simulations, as the meteorology (winds and
temperatures) in the GCM are synchronized with observed meteorology, direct
comparisons can be achieved by selecting model results that are collocated with
observations in space and time, and thus the model outputs can be evaluated in a more
rigorous manner.

In this study, we use the in-situ aircraft measurements from the NSF HIAPER

Pole-to-Pole Observations (HIPPO) campaign (Wofsy et al., 2011) to evaluate the



cloud properties simulated by the Community Atmosphere Model version 5 (CAM5).
During the HIPPO campaign, high-resolution (~230 m, 1Hz) and comprehensive
measurements of ambient environmental conditions (such as air temperature, pressure
and wind speed), cloud ice crystals and droplets were obtained. HIPPO also provides
a nearly pole-to-pole spatial coverage and relatively long flight hours (~400 hours in
total) in various seasons, making it a valuable dataset for GCM evaluations. To
facilitate the evaluation, CAM5 is run with specified dynamics where the model
meteorological fields (horizontal winds (U, V) and temperature (T)) are nudged
towards the NASA GEOS-5 analysis, while water vapor, cloud hydrometeors and
aerosols are calculated interactively by the model (Larmarque et al., 2012). Moreover,
we select collocated CAM5 output along the HIPPO aircraft flight tracks, and
compare the model simulations and observations directly. Our comparisons focus on
cloud occurrence, and cloud microphysical properties (e.g., ice water content, number
concentration and size distribution of ice particles) with a specific focus on cirrus
clouds. We also investigate the sensitivities of model simulated cirrus cloud properties
to the ice microphysics parameterizations as well as to the large scale forcing
associated with the nudging strategy.

The remainder of the paper is organized as follows. In section 2, we introduce the

HIPPO observational dataset and instrumentations. The model simulations and
experimental design are described in section 3. In section 4, we examine the model
performance in simulating cirrus cloud occurrence and microphysical properties and
investigate the reasons behind the model biases. Sensitivities of model results to





different nudging strategies are presented in section 5, and discussions and
conclusions in section 6.

**2    HIPPO aircraft observations**

The NSF HIPPO Global campaign provided comprehensive observations of
clouds and aerosols from 87°N to 67°S over the Pacific region during 2009 to 2011
(Wofsy et al., 2011). Observations were acquired using the National Science
Foundation's Gulfstream V (GV) research aircraft operated by the National Center for
Atmospheric Research (NCAR). During this three-year period, five HIPPO
deployments were carried out, with each deployment lasting from 23 days to about
one month. In total, the HIPPO campaign included 64 flights, 787 vertical profiles
(from the surface to up to 14 km), and 434 hours of high-rate measurements
(http://hippo.ucar.edu). In this study, we use the 1-Hz in-situ measurements of water
vapor, temperature, number concentration and size distribution of ice crystals as well
as the number concentration of cloud liquid droplets from HIPPO#2-5. HIPPO#1 did
not have ice probes onboard.
Water vapor was measured by the 25 Hz, open-path Vertical Cavity Surface
Emitting Laser (VCSEL) hygrometer (Zondlo et al., 2010). The accuracy and
precision of water vapor measurements was ~6% and ≤ 1%, respectively.
Temperature (T) was recorded by the Rosemount temperature probe. The accuracy
and precision of T measurements was 0.5 K and 0.01 K, respectively. Here saturation
vapor pressure is calculated following Murphy and Koop (2005), who stated that all





the commonly used expressions for the saturation vapor pressure over ice are within 1%
in the range between 170 and 273 K. Then we calculate relative humidity (RH) using
the saturation vapor pressure with respect to water (T>0°C) or with respect to ice
(T≤0°C). Unless explicitly stated otherwise, we refer to RH with respect to water
when T>0°C and RH with respect to ice when T≤0°C.

Ice crystal concentrations were measured by the two-dimensional cloud particle

imaging (2DC) ice probe (Korolev et al., 2011). The 2DC measures ice crystals with a
64-diode laser array at 25 μm resolution and the corresponding size range of 25 –
1600 μm. Outside this range, ice crystals between 1600 μm and 3200 μm are
mathematically reconstructed. A quality control was further applied to filter out the
particles with sizes below 75 μm in order to minimize the shattering effect and optical
uncertainties associated with 2DC data. Thus the number concentration ($N_i$) of ice
crystals with diameter from 75 μm to 3200 μm (binned by 25 μm) was derived and is
used here for model comparisons. The ice water content (IWC) is derived by
integrating the ice crystal mass at each size bin. Mass is calculated from diameter and
$N_i$ using the mass-dimension (m-D) relationship of Brown and Francis (1995). For the
ice crystal size distribution, a gamma function is assumed as in CAM5 (Morrison and
Gettelman, 2008):

$$\phi(D) = N_0 D^\mu \exp(-\lambda D) \tag{1}$$

where $D$ is diameter, $N_0$ is the intercept parameter, $\mu$ is the shape parameter which is
set to 0 currently, and $\lambda$ is the slope parameter. The slope and intercept for the
observed ice crystal size distributions are obtained by fitting Eq. (1) using the least



squares method as described in Heymsfield et al. (2008). Observed size distributions
that provided less than five bins of non-zero concentrations are not considered in
order to maintain a reasonable fit, which is similar to what was done in Eidhammer et
al. (2014). This removes about 8% of the total 1-Hz observations of ice clouds
(T≤-40°C). Furthermore, we only retain those fitted size distributions that are well
correlated with the measured ones, i.e., with a correlation coefficient larger than 0.6,
which leads to a further removal of 10% of the total 1-Hz ice crystal measurements.
Note that these screenings are applied only for the derivation of the slope and
intercept parameters for the ice crystal size distribution.

The cloud droplet number concentration ($N_d$) was measured by the Cloud Droplet

Probe (CDP) during the HIPPO campaign. The CDP measurement range of cloud
droplet diameter is 2-50 μm. Because 2DC and CDP probes may report both ice
crystals and liquid droplets, we adopted a rigorous criteria for the detection of clouds
in different temperature ranges. 99% of the observed $N_i$ are greater than 0.1 $L^{-1}$, thus a
threshold of 0.1 $L^{-1}$ is used to define in-cloud conditions. For T≤-40°C, we use the
criterion of $N_i$>0.1 $L^{-1}$ to detect the occurrence of ice clouds; For T>-40°C, the
occurrence of clouds including mixed-phase clouds (-40°C<T≤0°C) and warm
clouds (T>0°C) are defined by the conditions of either $N_i$>0.1 $L^{-1}$ or $N_d$>1 $cm^{-3}$. Here,
we only analyze CDP measurements with $N_d$>1 $cm^{-3}$ to avoid measurement noise as
determined by the sensitivity of the instrument.

The HIPPO dataset has been previously used for statistical analyses of ice cloud

formation conditions and microphysical properties, such as the conditions of the



birthplaces of ice clouds – the ice supersaturated regions, the evolutionary trend of
RH and $N_i$ inside cirrus clouds, and hemispheric differences in these cloud properties
(Diao et al., 2013; 2014a, b). In this study, we will use these observations to evaluate
CAM5 simulation of ice clouds. We use 10-second averaged measurements (~2.3 km
horizontal resolution) which are derived from 1 Hz (~230 m horizontal resolution)
observations. Although variations are found (mostly within a factor of 2 and
sometimes up to 2-3 for $N_i$, IWC and $\lambda$) within 10-second intervals, the 10-second
averaged observations shown in this study are similar to those based on 1-second
measurements.

**3    Model and experiment design**
**3.1  Model**

This study uses version 5.3 of CAM5 (Neale et al., 2012), the atmospheric

component of NCAR Community Earth System Model (CESM). The cloud
macrophysics scheme in CAM5 provides an integrated framework for treatment of
cloud processes and imposes full consistency between cloud fraction and cloud
condensates (Park et al., 2014). Deep cumulus, shallow cumulus, and stratus clouds
are assumed to be horizontally distributed in each grid layer without overlapping with
each other. Liquid stratus and ice stratus are assumed to have a maximum horizontal
overlap with each other. Stratiform microphysical processes are represented by a
two-moment cloud microphysics scheme (Morrison and Gettelman et al., 2008;
hereafter as version 1 of MG scheme (MG1)). MG1 was improved by Gettleman et al.



(2010) to allow the ice supersaturation. It is coupled with a modal aerosol model
(MAM, Liu et al. (2012a)) for aerosol-cloud interactions. Cloud droplets can form via
the activation of aerosols (Abdul-Razzak and Ghan, 2000). Ice crystals can form via
the homogeneous nucleation of sulfate aerosol, and/or heterogeneous nucleation of
dust aerosol (Liu and Penner, 2005; Liu et al., 2007). The moist turbulence scheme is
based on Bretherton and Park (2009). Shallow convection is parameterized following
Park and Bretherton (2009), and deep convection is treated following Zhang and
McFarlane (1995) with further modifications by Richter and Rasch (2008).

Compared to the default version 5.3, the CAM5.3 version we use includes version

2 of the MG scheme (MG2) as described by Gettelman and Morrison (2015) and
Gettelman et al. (2015). MG2 added prognostic precipitation (i.e., rain and snow) as
compared with the diagnostic precipitation in MG1. Note that current version of MG
scheme treats cloud ice and snow as different categories with their number and mass
predicted, respectively (Morrison and Gettelman, 2008). To be consistent with the
observations, here the number and mass concentrations of cloud ice and snow are
combined together to get the slope parameter $\lambda$ following Eidhammer et al. (2014).
**3.2 Experimental design for model-observation comparisons**

Model experiments are performed using specified dynamics, that is, online

calculated meteorological fields (U, V, and T) are nudged towards the GEOS-5
analysis (the control experiment, referred to as CTL hereafter), while water vapor,
hydrometeors and aerosols are calculated online by the model itself (Larmarque et al.,
2012). We also conduct two experiments where only U and V are nudged (referred to



as NUG_UV) and with nudging U, V, T and water vapor (Q) (referred to as
NUG_UVTQ). These results will be discussed in section 5. The model horizontal and
vertical resolutions are 1.9° × 2.5° and 56 vertical levels, respectively. The time step
is 30 min. The critical threshold diameter for autoconversion of cloud ice to snow ($D_{cs}$)
was found to be an important parameter affecting ice cloud microphysics (e.g., Zhang
et al., 2013; Eidhammer et al., 2014). $D_{cs}$ is set to 150 μm in MG2. We also conduct
two sensitive experiments using a value of 75 μm (referred to as DCS75) and 300 μm
(referred to as DCS300) for $D_{cs}$ (Table 1).

In the standard CAM5 model, homogeneous nucleation takes place on sulfate

aerosol in the Aitken mode with diameter greater than 0.1 μm (Gettelman et al., 2010).
We conduct a sensitivity experiment (referred to as SUL) by removing this size limit
(i.e., using all sulfate aerosol particles in the Aitken mode for homogeneous
nucleation). Recently, Shi et al. (2015) incorporated the effects of pre-existing ice
crystals on ice nucleation in CAM5, simultaneously removing the lower limit of
sulfate aerosol size and the upper limit of the sub-grid updraft velocity used for the ice
nucleation parameterization. Here a sensitivity experiment (referred to as PRE-ICE)
with the Shi et al. (2015) modifications is conducted (Table 1).

We run the model from June 2008 to December 2011 (i.e., 43 months) with the

first seven months as the model spin-up. For direct comparisons between model
results and observations, only model output collocated with HIPPO aircraft flights are
recorded. That is, we locate the model grid boxes in which the HIPPO aircraft was
transecting through, and then output the model results of these grid boxes at the





292 closest time stamps with respect to the flight time. In total, we have 130,577 in-situ

293 observation samples at 10-second resolution (~363 hours) for HIPPO#2-5. We note

294 that because the current CAM5 model cannot explicitly resolve the spatio-temporal

295 variability of dynamic fields and cloud properties inside a model grid box, there are

296 inevitably certain caveats in its comparison with in-situ observations. For example, as

297 the model time step is 30 min and horizontal grid spacing is ~200 km, there may be

298 cases where tens to hundreds of flight samples are located within one grid box at a

299 specific time stamp. In this study, we find that there are 1 to 170 observation samples

300 within a model grid box. Therefore, we may over-sample the model results within a

301 model grid box with multiple aircraft samples. However, we note that because of the

302 specific flight plan of the HIPPO campaign, most of the HIPPO flights were designed

303 to follow a nearly constant direction when flying from one location to the next, and

304 one vertical profile was generally achieved by about every 3 latitudinal degrees. This

305 unique flight pattern combined with the comparatively long flight hours helps to

306 provide a large amount of observation samples transecting through various climate

307 model grid boxes. In total, 635 model grid boxes are used in the direct comparisons

308 with observations. Considering that the actual horizontal area fraction of a model grid

309 box that the aircraft transected through is relatively small, derivations of grid-scale

310 mean observations which can represent the realistic characteristics for the whole grid

311 box are not possible. Nevertheless, we also derive the mean of observations within a

312 model grid box and compare them with model simulations, and the comparison results

313 are similar to those shown in Section 4. Note that vertical interpolation is taken to



account for the altitude variation of model variables for the direct comparison with
aircraft observations.

## 4    Results

### 4.1  Cloud occurrence

In this section, we will first demonstrate the model performance in simulating the
spatial distributions of clouds with a case study. Then we will show the overall
features of cloud occurrence for all comparison samples. To identify the reasons for
the model-observation discrepancies, we will analyze the meteorology conditions (e.g.,
T, Q and RH) and physics processes associated with the formation of clouds. The
probability density function (PDF) of ice supersaturation at clear-sky and inside ice
clouds will be examined.

### 4.1.1    Case study – a specific cloud system

During HIPPO deployment #4 and research flight 05, the GV aircraft flew from
the Cook Islands to New Zealand over the South Pacific Ocean on June 25–26, 2011
(Figure 1). Low-level clouds existed along almost all the flight tracks at 700–1000
hPa, and most of them were warm clouds (T>0°C). Mid-level (at 400–700 hPa) and
high-level clouds (at 250–400 hPa) were also observed. Generally the model captures
well the locations of cloud systems along the flight tracks on June 25, 2011. The
simulated ice clouds are located above liquid clouds and extend for thousands of
kilometers, which corresponds with the observed mid- to high-level clouds at



250–600 hPa at UTC 2200–2400 on June 25, 2011. However, the model misses the
low-level clouds observed on late June 25 and early June 26, and simulates a smaller
horizontal extent for the mid-level cloud at UTC 0230 on June 26. Overall, the
observed clouds on June 26 (further South) were more scattered than those on June 25.
The model is less capable of reproducing these scattered clouds. CAM5 is better able
to simulate cloud systems with larger spatial extents, since these systems are
controlled by the nudged large-scale meteorology.
Figure 2 shows the time series of RH, Q and T during the flight segment shown in
Figure 1. The observations show large spatial variability in RH even during the
horizontal flights on June 26. Overall, the simulated RH is within the range of the
observations but the model is unable to simulate the larger variability, which occurred
on sub-grid spatial scales. Both observed and simulated RH values are above 100%
when the model captures the clouds successfully at UTC 2240-2250 and 2310-2330
on June 25 and at UTC 0000-0010 on June 26 (denoted by green vertical bars),
although the simulated maximum grid-mean RH value is around 110%, which is
10-30% less than observed RH values. However, the model cannot capture some of
the observed clouds with large RH values within the grid boxes. For example, the
model misses the RH associated with low-level clouds (Figure 1) at UTC 2250-2310
when simulated grid-mean RH values are around 90% compared to observed values
of around 100%. Note that since the aircraft sampled only portions of the model grid
boxes, the "over-production" of cloud occurrences by the model shown in Figure 2
(blue vertical bars) may not necessarily be the case. Thus we will focus on the cases



when the model captures or misses the observed clouds within the model grid boxes.

The spatial distributions of RH play an important role in determining whether

modeled clouds occur at the same times and locations as those observed. Biases in
either Q or T may lead to discrepancies in RH (Figs. 2d and 2f). For example, at
around UTC 2150 on June 25, higher RH in the model is caused by the larger
simulated Q; at UTC 2250 on June 25, simulated lower RH is mainly caused by the
warmer T. To illustrate whether T or Q biases are the main cause for the RH biases,
we calculate the offline distribution of RH by replacing the modeled Q or T with the
aircraft observations, as shown in Figures 3a and 3b, respectively. After adopting the
observed T spatial distributions, the updated RH still misses the RH variability around
UTC 0230 – 0400 on June 26, while by adopting the observed Q spatial distribution,
the updated RH distribution is very close to the observed one. Thus, in this case study
the lack of a large RH spatial variability shown in the observations mainly results
from the model's lack of sub-grid scale variability of Q rather than that of T.
**4.1.2    Synthesized analyses on cloud occurrences and cloud fraction**

The overall performance of the model in simulating the cloud occurrences for all

flights in HIPPO 2–5 is shown in Table 2. In the model, clouds often occupy a
fraction of a grid box, and cloud fraction together with in-cloud liquid/ice number
concentrations are used to represent the occurrence of stratus clouds (Park et al.,
2014). For HIPPO, the occurrence of clouds is derived by combining the observations
of both liquid and ice number concentrations as described in section 2. In total, the
model captures 79.8% of observed cloud occurrences inside model grid boxes. For





different cloud types, the model reproduces the highest fraction (94.3%) of observed
ice clouds, and the second highest fraction (86.1%) for mixed-phase clouds. In
contrast, the model captures only about half (49.9%) of observed warm clouds. As
depicted in the case study in section 4.1.1, the missing of cloud occurrences are
mainly due to the insufficient representation of sub-grid variability of RH in the
model. Next we will further quantify the contribution of sub-grid water vapor and
temperature variations to sub-grid variability of RH.
**4.1.3    Decomposition of relative humidity biases**
The formation of liquid droplets/ice crystals depends on dynamical and
thermodynamical conditions such as temperature, water vapor and updraft velocity
(Abdul-Razzak and Ghan, 2000; Liu et al., 2007, 2012b; Gettelman et al., 2010). The
fraction of liquid/ice stratus clouds is calculated empirically from the grid-mean RH
(Park et al., 2014). Thus RH is an important factor for both model representations of
cloud occurrences and cloud fraction. RH is a function of pressure, temperature and
water vapor. Since we only compare observations with the simulation results on the
same pressure levels, differences of RH ($d$RH) between simulations and observations
(i.e., model biases in RH) only result from the differences in temperature and water
vapor. We calculate the contributions of biases in water vapor and temperature to the
biases in RH following the method that was used to analyze RH spatial variability in
Diao et al. (2014a). $RH_o$ (observations) and $RH_m$ (model results) are calculated as:
$$RH_m = \frac{e_m}{e_{s,m}}, \; RH_o = \frac{e_o}{e_{s,o}} \qquad\qquad (2)$$
where $e_o$ and $e_m$ are observed and simulated water vapor partial pressure, respectively,





and $e_{s,o}$ and $e_{s,m}$ are observed and simulated saturation vapor pressure over ice (T≤0°C)
or over water (T>0°C) in the observations or the model, respectively.
Here $d$RH is calculated from the difference of simulated grid-mean RH (with
vertical variances taken into account by the vertical interpolation) and in-situ
observations. We define $de = (e_m - e_o)$, and $d(\frac{1}{e_s}) = \frac{1}{e_{s,m}} - \frac{1}{e_{s,o}}$, therefore $d$RH is
$$dRH = RH_m - RH_o = de \cdot \frac{1}{e_{s,o}} + e_o \cdot d(\frac{1}{e_s}) + de \cdot d(\frac{1}{e_s}) \qquad (3)$$

Thus $d$RH can be separated into three terms: the first term is the contribution from the
water vapor partial pressure ($d$RH$_q$), the second term from temperature ($d$RH$_T$), and
the third term for concurrent impact of biases in temperature and water vapor
($d$RH$_{q,T}$).
Figure 4 shows the contributions of these three terms to $d$RH for different
temperature ranges. All the three terms as well as $d$RH are given in percentage. The
intercepts and slopes of linear regression lines for $d$RH$_q$ versus $d$RH, $d$RH$_T$ versus
$d$RH, and $d$RH$_{T,q}$ versus $d$RH are also presented. As temperature is constrained by
GEOS-5 analysis, the bias in temperature is reduced (although not eliminated) to
mostly within ±7°C. A considerable amount of discrepancy in RH exist between
model and observations. The model successfully captures the clouds (green symbols)
when the simulated RH is close to observations in all the three temperature ranges.
The model tends to miss the clouds (red symbols) when lower RH is simulated, and
produces spurious clouds (blue symbols) when higher RH is simulated. Regarding the
contributions of $d$RH$_q$ and $d$RH$_T$ to $d$RH, the slopes of the linear regression for $d$RH$_q$




423 versus $d$RH are 0.748, 0.933 and 0.786 for T≤-40°C, -40°C<T≤0°C and T>0°C,

424 respectively, which are much larger than those for $d$RH$_T$ versus $d$RH (0.087, 0.072

425 and 0.210 for the three temperature ranges, respectively). This indicates that most of

426 the biases in RH are contributed by the biases in water vapor ($d$RH$_q$). However, for

427 T>0°C, although $d$RH$_q$ still dominates, $d$RH$_T$ contributes notably to 21% of the RH

428 biases. For T≤-40°C, $d$RH$_{q,T}$ also contributes about 17% to $d$RH, indicating

429 concurrent impact from biases of T and water vapor. In contrast, for -40°C<T≤0°C

430 and T>0°C, the contributions of $d$RH$_{q,T}$ to $d$RH are negligible. We note that the slopes

431 of linear regression lines for $d$RH$_q$ versus $d$RH and $d$RH$_T$ versus $d$RH indicate the

432 average contributions from water vapor and temperature biases to the RH biases,

433 respectively. The values of $d$RH$_T$ can occasionally reach up to ±100%, which

434 suggests the large impact from temperature biases in these cases. In addition, the

435 $d$RH$_T$ and $d$RH$_q$ terms can have the same (opposite) signs, which would lead to larger

436 (lower) total biases in RH. The coefficients of determination, $R^2$, for the linear

437 regressions indicate that $d$RH$_q$ versus $d$RH has a much stronger correlation than that

438 of $d$RH$_T$ versus $d$RH.

439 **4.1.4 Ice supersaturation**

440  Ice nucleation only occurs in the regions where ice supersaturation exists.

441 Different magnitudes of ice supersaturation are required to initiate homogeneous and

442 heterogeneous nucleation (Liu and Penner, 2005). The distribution of ice

443 supersaturation may provide insights into the mechanisms for ice crystal formation

444 (e.g., Haag et al., 2003). In CAM5, ice supersaturation is allowed (Gettelman et al.,





2010). Homogeneous nucleation occurs when T≤-35°C and ice supersaturation
reaches a threshold ranging from 145% to 175%. Dust aerosol can serve as INPs
when RH>120%. Ice supersaturation will be relaxed back to saturation via the vapor
deposition process (Liu et al., 2007; Gettelman et al., 2010).

To examine the discrepancies in ice supersaturation between model results and

observations, we compare the distribution of RH for conditions in clear-sky and
within cirrus clouds (Figure 5). The analysis is limited to the conditions of T≤-40°C
for both model simulations and observations. In CAM5, RH diagnosed in different
sections of the time integration procedure can be different due to the time splitting
algorithm. We present here both the RH before and after the microphysical processes.

The observations show that ice supersaturation exists in both clear-sky and

inside-cirrus conditions. In clear-sky environments, the PDF of RH shows a
continuous decrease with RH values in subsaturated conditions, followed by a
quasi-exponential decrease with the RH above saturation. The maximum RHi reaches
up to 150%. In cirrus clouds, most of RH values range from 50% to 150% with a peak
in the PDF near 100%. This feature is consistent with the results of Diao et al. (2014b),
who used 1-second HIPPO measurements and separated the southern and the northern
hemispheres for comparison.

The PDFs of modeled RH before and after the microphysical processes are very

similar except the latter one has slightly lower probability of RHi above 140% for
inside-cirrus conditions. Compared to the observations, the model can simulate the
occurrences of ice supersaturation in both clear-sky and in-cloud conditions. However,





inside cirrus clouds, the simulated PDF of RH peaks around 120% instead of 100% as
observed. Outside the cirrus clouds (clear-sky), the model simulates a much lower
probability of ice supersaturation with the maximum RH value around 120%. The
largest ice supersaturation simulated by CAM5 under clear-sky conditions is around
20%, which corresponds to the ice supersaturation of 20% assumed in the model for
the activation of heterogeneous nucleation. This indicates the dominant mode of
heterogeneous nucleation in the model. However, the observations show much higher
frequencies of ice supersaturations larger than 20%, indicating higher RH thresholds
for homogeneous nucleation or heterogeneous nucleation.

**4.2 Microphysical properties of ice clouds**
Together with cirrus cloud fraction, the ice crystal number concentration and size
distribution within cirrus clouds determine the radiative forcing of cirrus clouds. In
this section, we will present the evaluation of modeled microphysical properties of
cirrus clouds for T≤-40°C. As measurements of ice crystal number concentration
include both ice and snow crystals, for comparison with observations, we combine the
cloud ice and snow simulated in the model (hereafter referred as ice crystals).
Following Eidhammer et al. (2014), the slope and intercept parameters of the gamma
function for the ice crystal size distribution simulated by the model are derived from
the total number concentration and mass mixing ratio of cloud ice and snow, which
are the integrations of the first and third moments of the size distribution function.
The simulated number concentration of ice crystals with sizes larger than 75 μm is



calculated by the integration of gamma size distributions from 75 μm to infinity. The
simulated IWC for ice crystals with sizes larger than 75 μm is also derived by
integrating the mass concentration of cloud ice and snow from 75 μm to infinity. We
note that about 94% of total cirrus cloud samples are at temperatures between -60°C
and -40°C.

**4.2.1    Ice crystal size distribution**


Direct comparison of the slope parameter ($\lambda$) for ice crystal size distributions is

shown in Figure 6. The slope parameter λ determines the decay rate of a gamma
function in relation to the increasing diameter. With a larger λ, the decay of a gamma
function with increasing size is faster and there are relatively fewer large ice crystals.
The number-weighted mean diameter can be defined as the inverse of λ (i.e., $\lambda^{-1}$). As
shown in Figure 6, the observed λ is generally within the range from $10^3$ to $10^5$ $m^{-1}$.
The model reproduces the magnitude of λ for some of the observations, but tends to
overestimate the observations for smaller λ values ($10^3$ to $10^4$ $m^{-1}$). This indicates that
the model produces higher fractions of ice crystals at smaller sizes, and the
number-weighted mean diameter is underestimated. Moreover, the model generally
simulates λ in a narrower range of $7.5 \times 10^3$ to $7 \times 10^4$ $m^{-1}$ for the three experiments with
different $D_{cs}$ (CTL, DCS75, DCS300). SUL and PRE-ICE simulate a wider range of λ
which is comparable to the observations but tends to shift λ to larger values ($5 \times 10^4$ to
$1 \times 10^5$ $m^{-1}$). All the experiments rarely simulated the occurrence of small λ (below
$7.5 \times 10^3$ $m^{-1}$).

Figure 7 shows the relationship of λ with temperature from observations and



model simulations. Here, both the geometric means and the standard deviations of λ
for each temperature interval of 4°C are also shown. Although the observed λ doesn't
monotonically decrease with increasing temperature, overall an decreasing trend can
be found for the whole temperature range below -40°C. This indicates a general
increase in the number-weighted mean diameter of ice crystals with increasing
temperature. The correlation between λ and temperature from HIPPO is similar to that
from the Atmospheric Radiation Measurements Spring Cloud Intensive Operational
Period in 2000 (ARM-IOP) and the Tropical Composition, Cloud and Climate
Coupling (TC4) campaigns as shown in Eidhammer et al. (2014), but the HIPPO
observations extend to lower temperatures than ARM-IOP and TC4 observations
where temperatures are mostly above -56 °C. In addition, HIPPO observations show a
broader scatter range of λ, which may be because HIPPO sampled ice crystals at
various environment conditions as the flight tracks covered much wider areas and
lasted for much longer periods. The decrease of λ with increasing temperature has
been shown in many other studies (e.g., Heymsfield et al., 2008; 2013). Such a feature
is mainly due to more large ice particles at higher temperatures, and can also be partly
explained by more ice crystals formed from nucleation and less water vapor available
for ice crystal growth at lower temperatures (Eidhammer et al., 2014).

Compared to the observations, the simulated mean λ is about 2-4 times larger for

all the experiments, indicating that the model simulates smaller mean sizes for ice
crystals. The simulated λ decreases with increasing temperature, which is generally
consistent with the observations. In addition, the geometric standard deviations (less





than 2) of simulated λ are smaller than observed (around 2-3). This can be partly
explained by the fact that in-situ observations sampled the sub-grid variability of
cloud properties.

The difference of simulated λ is within a factor of 2 among the five experiments

when temperature is between -40°C and -56 °C, and is larger (around 2-4) when
temperature is below -56 °C. For the experiments with different $D_{cs}$, CTL and DCS75
simulated λ are close to each other when temperature is between -40°C and -60 °C,
and DCS300 simulates larger λ compared to DCS75 and CNTL. For temperatures
between -64°C and -72 °C, CTL and DCS300 simulated λ are close to each other and
both are larger than that of DCS75. For the experiments with different ice nucleation
parameterizations, both SUL and PRE-ICE simulate larger λ than CTL especially for
temperatures below -56 °C. SUL simulates the largest λ of all the experiments. This
can be explained by much larger number concentration of ice crystals (for all size
range, figure not shown) simulated by SUL, while IWC is not very different from
other experiments (section 4.2.3).

**4.2.2    Ice crystal number concentration**

Figure 8 shows the comparison of in-cloud number concentrations ($N_i$) of ice

crystals with diameters larger than 75 μm between observations and simulations. The
magnitude of observed $N_i$ varies by three orders of magnitude from $10^{-1}$ L$^{-1}$ to $10^2$ L$^{-1}$.
The model simulates reasonably well the range of $N_i$ in cirrus clouds. However, the
model tends to underestimate $N_i$ for all the experiments except DCS75. About 13%





(DCS75) to 30% (PRE) of observations are underestimated in the model by a factor of
10. The underestimation of $N_i$ may be partly attributed to the fact that the model
underestimates the ice crystal size (section 4.2.1), leading to a smaller fraction of ice
crystals with diameter larger than 75 μm. Additional bias may result from the bias in
the total ice crystal number concentration, although the observations are not available
for comparison. We also compare simulated $N_i$ with observed in-cloud $N_i$ averaged
within the model grid boxes. We choose the flight segments with over 300 1-second
aircraft measurements within an individual model grid and calculate the average for
in-cloud $N_i$ of ice clouds (T≤-40 °C). The comparison results are, however, similar to
those shown in Figure 8.

DCS75 reasonably simulates the occurrence frequency of $N_i < 1$ L$^{-1}$ albeit with

significantly higher frequency for $N_i$ around 1-5 L$^{-1}$ and lower frequency for $N_i$
around 5-10 L$^{-1}$. Most of the experiments cannot reproduce the occurrence frequency
of high $N_i$ ($N_i > 50$ L$^{-1}$) except DCS75 and PRE-ICE.

The relationships between $N_i$ and temperature are shown in Figure 9. Since $N_i$

here only takes into account of ice crystals larger than 75 μm, the geometric mean of
observed $N_i$ generally ranges between 5-10 L$^{-1}$ for temperatures below -40°C, which
is 1-2 orders of magnitude lower than the number of ice crystals between 0.3-775 μm
from observations complied by Krämer et al. (2009) and between 10-3000 μm from
the SPARTICUS campaign (Zhang et al., 2013), but is comparable to the number of
ice crystals in the same size range from the ARM-IOP and TC4 campaigns
(Eidhammer et al., 2014). The geometric standard deviation of observed $N_i$ within a




temperature interval of 4°C can be as high as a factor of 5.

The model simulates no apparent trends of $N_i$ when temperature decreases from

-40°C to -60°C for the experiments CTL, DCS75 and PRE-ICE. The model simulates
somehow larger $N_i$ with decreasing temperatures for the experiments DCS300 and
SUL. Increase of $N_i$ at lower temperatures in SUL may indicate the occurrence of
homogeneous nucleation. Overall, simulated $N_i$ is sensitive to $D_{cs}$. Simulated $N_i$ is
also sensitive to the number of sulfate aerosol particles for homogeneous nucleation.
With the removal of the lower size limit (0.1 μm diameter) of sulfate aerosol particles
for homogeneous nucleation in the experiment SUL, simulated $N_i$ is significantly
higher than that in CTL. This result is consistent with that of Wang et al. (2014).

Although some experiments can simulate a similar magnitude of $N_i$ as the

observations in some temperature ranges, most of the experiments underestimate $N_i$
and some experiments (CTL and PRE-ICE) underestimate $N_i$ for all the temperature
ranges. Overall DCS75 simulates the closest magnitude of $N_i$ with the observations
for temperatures from -40°C to -64°C.

**4.2.3    Ice water content**

Figure 10 shows the comparison of in-cloud IWC for ice crystals with diameter

larger than 75 μm between observations and simulations. The magnitude of observed
IWC varies by four orders of magnitude from $10^{-2}$ to $10^2$ mg m$^{-3}$, which is within the
range of observed IWC in previous studies (Kramer et al., 2016; Luebke et al., 2016).
Observed IWC here is mostly larger than 1 mg m$^{-3}$. Compared to the observations, the





model for all the experiments underestimates observed IWC for 70%-95% of the
samples and by one order of magnitude for 25%-45% of the samples. Although the
model reproduces the highest occurrence frequency of IWC around 1-5 mg m$^{-3}$, the
model simulates more occurrence of IWC below 1 mg m$^{-3}$ and fewer occurrence of
IWC above 5 mg m$^{-3}$.

The relationships between IWC and temperature are shown in Figure 11. An

overall increasing trend of observed IWC with temperature is found for the entire
temperature range. The observed relationship between IWC and temperature is
consistent with those shown in the previous studies (e.g., Kramer et al., 2016; Luebke
et al., 2016). However, the mean IWC from HIPPO is 3-5 times as large as previous
observations (Kramer et al., 2016; Luebke et al., 2016). Observations here only
account for ice crystals with diameter larger than 75 μm and thus it is less frequent
that observed IWC is lower than 1 mg m$^{-3}$. In contrast, previous studies showed that
IWC (including smaller sizes of ice crystals) lower than 1 mg m$^{-3}$ was often measured
in observations. This contributes to the mean IWC shown here being larger than that
in the previous studies.

The simulated IWC is lower than observations for all the experiments at

temperatures between -40°C and -60 °C where most of the observations were made.
The model also simulates less variation of IWC with temperature when temperature is
between -40°C and -60 °C. When temperature is below -60 °C, a steep decrease of
IWC is found in some experiments (e.g., CTL, SUL). Considering the large scatter of
IWC and relatively few samples available, this may be due to a lack of a sufficient





number of samples. Therefore, more observations are needed to have a robust
comparison for relatively low temperatures (i.e., temperatures below -60 °C).
Simulated IWC is more sensitive to $D_{cs}$ than to ice nucleation.

**5    Impact of Nudging**

In previous sections, we have nudged the simulated winds and temperature

towards the GEOS5 analysis, but kept the water vapor on-line calculated by the model
itself. We showed that the model captures a large portion (79.8%) of cloud
occurrences presented in the observations. We also identified the RH bias in the
simulation and attributed the RH bias mainly to the bias in water vapor. As the bias in
temperature is reduced in the nudging run compared to the free run, the attribution of
RH bias in the free-running model (i.e., no nudging applied) is still unclear. To
examine the impact of nudging strategies on the cloud occurrences and the attribution
of RH bias, we conducted two additional experiments: one with neither temperature
nor specific humidity nudged to the analysis (hereafter referred as NUG_UV), and the
other one with both temperature and specific humidity nudged to the analysis
(hereafter referred as NUG_UVTQ). Without nudging temperature, the model
experiment (NUG_UV) has a cold temperature bias of -1.8ºC on average relative to
the HIPPO observations (Figure not shown). In comparison, the temperatures
simulated by CTL and NUG_UVTQ are more consistent with in situ aircraft
observations, and the mean temperature is slightly underestimated by 0.22 ºC and
0.28 ºC in these two experiments, respectively. By nudging specific humidity, the





model experiment (NUG_UVTQ) improves the simulation of grid-mean water vapor
concentrations by eliminating the biases especially for the cases with low water vapor
concentrations (less than 20 ppmv, Figure not shown). NUG_UV captures 86.0%,
80.9%, and 39.7% of observed ice, mixed-phase, and warm clouds, respectively,
which are slightly smaller than those of CTL (i.e., 94.3%, 86.1%, and 49.9%,
respectively). For NUG_UVTQ, although 73.5% of ice clouds are captured, the model
captures only 61.8% of mixed-phase clouds and 31.4% of warm clouds. The worse
simulation in NUG_UVTQ may be because the nudged water vapor is not internally
consistent with the modeled cloud physics, which deteriorates the simulation of cloud
occurrences.

As seen in Table 3, in the two new nudging experiments (NUG_UV and

NUG_UVTQ), modeled RH biases in the comparison with in-situ observations also
mainly result from the discrepancies of water vapor. The contribution of $d\mathrm{RH}_q$ to $d\mathrm{RH}$
ranges from 65.8% to 92.5%, which are slightly smaller than those in CTL. In
NUG_UV, as the model underestimates the temperature, modeled RH is
systematically higher than observations, especially for T≤-40ºC where the absolute
value of RH is overestimated by 30% on average. The large T bias leads to a smaller
contribution from the water vapor bias ($d\mathrm{RH}_q$) and a larger contribution from the
concurrent bias in temperature and water vapor ($d\mathrm{RH}_{q,T}$). When both T and Q are
nudged in NUG_UVTQ, the contributions of the three terms to $d\mathrm{RH}$ are generally
similar to those in CTL. A larger contribution from temperature ($d\mathrm{RH}_T$) is found for
temperature above 0ºC in NUG_UVTG. This may be a result of smaller contributions





from either $d\text{RH}_q$ or $d\text{RH}_{q,T}$ due to the reduced water vapor bias. We also examined
the in-cirrus microphysical properties simulated by these two new nudging
experiments. The model features such as underestimations of $N_i$, IWC, and mean ice
crystal size are similar to those in CTL and are not sensitive to the nudging strategy
used.

**6    Discussion and Conclusions**
In this study, we evaluated the macro- and microphysical properties of ice clouds
simulated by CAM5 using in-situ measurements from the HIPPO campaign. The
HIPPO campaign sampled over the Pacific region from 67°S to 87°N across several
seasons, making it distinctive from other previous campaigns and valuable for
providing insight into evaluating model performance. To eliminate the impact of
large-scale circulation biases on the simulated cloud processes, we ran CAM5 using
specified dynamics which nudge the simulated meteorology (U, V and T) towards the
GEOS-5 analysis while keeping water vapor, hydrometeors, and aerosols online
calculated by the model itself. Model results collocated with the flight tracks spatially
and temporally are directly compared with the observations. Modeled cloud
occurrences and in-cloud ice crystal properties are evaluated, and the reasons for the
biases are examined. We also examined the model sensitivity to $D_{cs}$ and different
parameterizations for ice nucleation.
The model can reasonably capture the vertical configuration and horizontal
extension of specific cloud systems. In total, the model captures 79.8% of observed



cloud occurrences within model grid boxes. For each cloud type, the model captures
94.3% of observed ice clouds, and 86.1% of mixed-phase and 49.9% of warm clouds.
This result is only modestly sensitive to whether meteorological fields (T and Q) are
nudged. The model cannot capture the large spatial variability of observed RH, which
is responsible for much of the model missing of low-level warm clouds. A large
portion of the RH bias results from the discrepancy in water vapor, with a small
portion from the discrepancy in temperature. The model also underestimates the
occurrence frequencies of ice supersaturation higher than 20% under clear-sky
conditions (i.e., outside of cirrus clouds), which may indicate too low threshold for
initiating heterogeneous ice nucleation in the model. In fact, a study comparing the
observed RH distributions with real-case simulations of the Weather Research and
Forecasting (WRF) model suggested that the threshold for initiating heterogeneous
nucleation should be set at RHi ≥ 125% (D'Alessandro et al., submitted).

Down to the micro-scale of cirrus clouds (T≤-40 ºC), the model captures well the

decreasing trend of λ with increasing temperature from -72 °C to -40°C. However, the
simulated λ values are about 2-4 times on average larger than observations at all the
4ºC temperature ranges for all the experiments with different $D_{cs}$ and different ice
nucleation parameterizations. This indicates that the model simulates a smaller mean
size of ice crystals in each temperature range. The model is mostly able reproduce the
magnitude of observed $N_i$ (to within one order of magnitude) for ice crystals with
diameter larger than 75 μm, yet generally underestimates $N_i$ except for the DCS75
simulation. Simulated $N_i$ is sensitive to $D_{cs}$ and the number of sulfate aerosol particles





for homogeneous nucleation used in the model. No apparent correlations between the
mean $N_i$ and temperature are found in the observations, while a decrease of $N_i$ with
increasing temperature is found in the two simulations (DCS300 and SUL). All the
experiments underestimate the magnitude of IWC for ice crystals larger than 75 μm.
The observations show an overall decreasing trend of IWC with decreasing
temperature while the model simulated trends are not as strong. Simulated IWC is
sensitive to $D_{cs}$ but less sensitive to the different parameterizations of ice nucleation
examined here.

Current climate models have typical horizontal resolutions of tens to hundreds of

kilometers and are unable to represent the large spatial variability of environmental
conditions for cloud formation and evolution within a model grid box. A previous
study of Diao et al. (2014a) shows that the spatial variability of water vapor
dominantly contribute to the spatial variability in RH, compared with the
contributions from those of temperature. Here our comparisons of model simulations
with observations show that the biases in water vapor spatial distributions are the
dominant sources of the model biases in RH spatial distributions. Thus it is a priority
to develop parameterizations that are able to treat the sub-grid variability of water
vapor for climate models. There are also substantial sub-grid variations of cloud
microphysical properties shown in previous observational studies (e.g., Lebsock et al.,
2013). Currently a framework for treating the sub-grid variability of temperature,
moisture and vertical velocity has been developed and implemented into CAM5
(Bogenschutz et al., 2013). A multi-scale modeling framework has also been



developed to explicitly resolve the cloud dynamics and cloud microphysics down to
the scales of cloud-resolving models (e.g., Wang et al., 2011; Zhang et al., 2014). The
PDFs of sub-grid scale distributions can be sampled on sub-columns for cloud
microphysics (Thayer-Calder et al., 2015). With the increase of model resolutions for
future model developments, the subgrid variablility of temperature, moisture, and
cloud microphysics and dynamics will be better resolved. We plan to evaluate the
model performances at higher resolutions.

Given the various environmental conditions and aerosol characteristics in the

atmosphere, the formation and evolution of ice crystals are not well understood, and it
is even more challenging for climate models to represent these processes. For the bulk
ice microphysics used in our model, several assumptions have to be made to simulate
both $N_i$ and $\lambda$. One of them is to partition the ice crystals into cloud ice and snow
categories, while using $D_{cs}$ to convert cloud ice to snow. Thus a more physical
treatment of ice crystal evolution such as using bin microphysics (e.g., Bardeen et al.,
2013; Khain et al., 2015) or a single category to represent all ice-phase hydrometeors
(Morrison and Milbrandt, 2015) is needed.

**Acknowledgements**
X. Liu and C. Wu acknowledge support of the U.S. Department of Energy's
Atmospheric System Research Program (grant DE-572 SC0014239). The authors
would like to acknowledge the use of computational resources (ark:/85065/d7wd3xhc)
at the NCAR-Wyoming Supercomputing Center provided by the National Science





Foundation and the State of Wyoming, and supported by NCAR's Computational and
Information Systems Laboratory. We appreciate the efforts of the National Center for
Atmospheric Research (NCAR) Earth Observing Laboratory flight, technical, and
mechanical crews during the National Science Foundation (NSF) HIPPO Global
campaign, in particular the PIs of the HIPPO Global campaign: S. Wofsy, R. Keeling,
and B. Stephens. NCAR is sponsored by NSF. We acknowledge the support of the
VCSEL hygrometer by M. Diao, M. Zondlo and S. Beaton, the support of 2DC probe
by A. Bansemer, C. J. Webster and D. C. Rogers. We also acknowledge the funding
of NSFAGS-1036275 for field support and data analyses from the VCSEL
hygrometer in the HIPPO Global campaign. M. Diao gratefully acknowledges the
support from the NCAR Advanced Study Program (ASP) postdoctoral fellowship in
Oct 2013– Aug 2015. Final data sets and documentation from the NSF HIPPO Global
campaign can be accessed at <http://hippo.ornl.gov> in the Carbon Dioxide
Information Analysis Center Data Archive at Oak Ridge National Laboratory. We
thank T. Eidhammer from NCAR for her help on model result analysis.

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





Table 1. CAM5 experiments

| Experiment name | Nudging | Ice microphysics parameterizations |
|---|---|---|
| CTL | U, V, T | Threshold diameter for autoconversion of cloud ice to snow ($D_{cs}$) set to 150 μm |
| DCS75 | U, V, T | As CTL, but with $D_{cs}$=75 μm |
| DCS300 | U, V, T | As CTL, but with $D_{cs}$=300 μm |
| SUL | U, V, T | As CTL, but without the lower limit (0.1 μm) for sulfate particle diameter for homogeneous freezing |
| PRE-ICE | U, V, T | As CTL, but with the impacts of pre-existing ice crystals on ice nucleation (Shi et al., 2015) |
| NUG_UV | U, V | As CTL |
| NUG_UVTQ | U, V, T, Q | As CTL |








1041 Table 2. The numbers of cloud occurrences in the 10-second averaged observations

1042 ($N_{obs}$), as well as those that CAM5 captures ($N_{cap}$) or misses ($N_{mis}$) the observed

1043 clouds within the model grid boxes for different temperature ranges. The ratio of $N_{cap}$

1044 and $N_{mis}$ to $N_{obs}$ are given in parenthesis next to them, respectively.

| Cloud type | Temperature ranges | $N_{obs}$ | $N_{cap}$ | $N_{mis}$ |
|---|---|---|---|---|
| Ice cloud | T≤-40°C | 3101 | 2925 (94.3%) | 176 (5.7%) |
| Mixed-phase cloud | -40°C<T≤0°C | 8768 | 7546 (86.1%) | 1222 (13.9%) |
| Warm cloud | T>0°C | 3334 | 1665 (49.9%) | 1669 (50.1%) |
| All | | 15203 | 12136 (79.8%) | 3067 (20.2%) |





Table 3. The intercepts and slopes of the regression lines (i.e., $Y=a+b*X$) for $d\mathrm{RH_q}$
versus $d\mathrm{RH}$, $d\mathrm{RH_T}$ versus $d\mathrm{RH}$, and $d\mathrm{RH_{q,T}}$ versus $d\mathrm{RH}$ in the three experiments CTL,
NUG_UV, and NUG_UVTQ, respectively. The coefficients are determination (i.e., $R^2$)
for each regression line are also presented.

| | | T≤-40°C | | | -40°C<T≤0°C | | | T>0°C | | |
|---|---|---|---|---|---|---|---|---|---|---|
| | | *a* | *b* | $R^2$ | *a* | *b* | $R^2$ | *a* | *b* | $R^2$ |
| CTL | $d\mathrm{RH_q}$ | 5.209 | 0.748 | 0.663 | 4.632 | 0.933 | 0.786 | 0.177 | 0.786 | 0.840 |
| | $d\mathrm{RH_T}$ | -0.798 | 0.087 | 0.071 | -3.013 | 0.072 | 0.039 | -0.706 | 0.210 | 0.262 |
| | $d\mathrm{RH_{q,T}}$ | -4.411 | 0.165 | 0.241 | -1.619 | -0.005 | .0004 | 0.529 | 0.004 | 0.001 |
| NUG_UV | $d\mathrm{RH_q}$ | -16.85 | 0.723 | 0.562 | -5.589 | 0.866 | 0.614 | -5.207 | 0.658 | 0.698 |
| | $d\mathrm{RH_T}$ | 29.96 | -0.103 | 0.024 | 10.09 | -0.013 | .0005 | 4.804 | 0.265 | 0.188 |
| | $d\mathrm{RH_{q,T}}$ | -13.11 | 0.380 | 0.487 | -4.498 | 0.148 | 0.088 | 0.402 | 0.078 | 0.085 |
| NUG_UVTQ | $d\mathrm{RH_q}$ | -2.851 | 0.813 | 0.770 | 2.260 | 0.925 | 0.672 | -1.773 | 0.733 | 0.761 |
| | $d\mathrm{RH_T}$ | 3.964 | 0.073 | 0.040 | -0.265 | 0.094 | 0.038 | 1.892 | 0.308 | 0.311 |
| | $d\mathrm{RH_{q,T}}$ | -1.113 | 0.114 | 0.262 | -1.996 | -0.019 | 0.003 | -0.119 | -0.041 | 0.095 |






**Figure captions:**
Figure 1. Cloud occurrences simulated by CAM5 (blue and green shaded areas)
compared with HIPPO observations (crosses) during HIPPO#4 Research Flight 05
(H4RF05) from Rarotonga, the Cook Islands (21.2°S, 159.77°W) to Christchurch,
New Zealand (43.48°S, 172.54°E) on June 25–26, 2011. Modeled in-cloud ice crystal
number concentration and cloud droplet number concentration are denoted by blue
and green shaded areas, respectively. Three temperature ranges are used to categorize
the combined measurements of 2DC and CDP probes. The criteria for defining
observed cloud occurrences are described in section 2.
Figure 2. Spatial variabilities of RH, water vapor (Q), and temperature (T) from
CAM5 simulation and HIPPO observation (left), and their differences (right).
Absolute difference between CAM5 and HIPPO is shown for RH and T, while the
ratio between CAM5 and HIPPO is shown for Q. Model performances are denoted by
shaded vertical bars: green (red) denotes when the model captures (misses) the
observed cloud occurrences, and blue denotes when the model simulates a cloud that
is not present in the observation.
Figure 3. As Figure 2a, but for RH recalculated by replacing the model output with
either (a) observed Q or (b) observed T values.
Figure 4. Corresponding (top) $d$RH$_q$ versus $d$RH, (middle) $d$RH$_T$ versus $d$RH, and
(bottom) $d$RH$_{q,T}$ versus $d$RH (unit: %) for different temperature ranges. The colors
indicating three types of model performances in simulating clouds as described in
Fig.2: green ("captured"), red ("missed") and blue ("overproduced"). The black lines
denote the linear regressions of the samples (i.e., $Y=a+b*X$), and the intercept (i.e., a)
and slope (i.e., b) of the regression lines as well as the coefficient of determination
(i.e., $R^2$) are shown in the legend.
Figure 5. Observed and simulated probability density functions (PDFs) of relative
humidity with respect to ice (RHi, unit: %) for T≤-40°C separated into clear-sky and
in-cirrus conditions. PDFs of RHi before and after cloud microphysics in the
simulations are both shown. The RHi is binned by 2% for the calculation of PDF. The
PDFs (when RHi>100%) follow an exponent decay: ln(PDF)=$a+b*$RHi. The values
of a and b for each PDF are also shown in dark red (observed), dark blue (simulated
before ice nucleaction), and dark green (simulated after cloud microphysics),
respectively. Note blue lines are mostly invisible as overlaid by green lines.





Figure 6. (a-e) Scatterplot of observed versus simulated slope parameter (λ) of the
gamma size distribution function for each experiments, and (f) the frequency of λ for
each range. Note that all the comparisons are restricted to the cases when the model
captures observed ice clouds (T≤-40 °C).
Figure 7. λ versus temperature from the measurements and simulations. The lines are
the geometric mean binned by 4°C, with the vertical bars denoting the geometric
standard deviation. Note that the comparisons are restricted to the cases when the
model captures the observed ice clouds (T≤-40 °C).
Figure 8. As Figure 6, but for the number concentrations ($N_i$) of ice crystals with
diameters larger than 75 μm for all the experiments. Note that both the comparisons
are restricted to the cases when the model captures observed ice clouds (T≤-40 °C).
Figure 9. As Figure 7, but for $N_i$.
Figure 10. As Figure 8, but for the comparison of ice water content (IWC).
Figure 11. As Figure 9, but for ice water content (IWC) versus temperature.






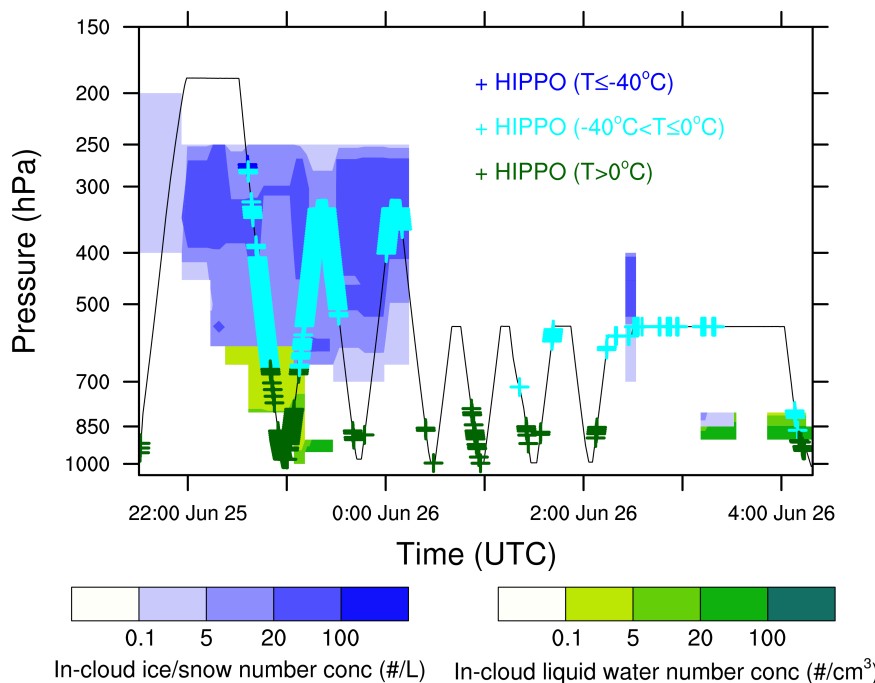


Figure 1. Cloud occurrences simulated by CAM5 (blue and green shaded areas)
compared with HIPPO observations (crosses) during HIPPO#4 Research Flight 05
(H4RF05) from Rarotonga, the Cook Islands (21.2°S, 159.77°W) to Christchurch,
New Zealand (43.48°S, 172.54°E) on June 25–26, 2011. Modeled in-cloud ice crystal
number concentration and cloud droplet number concentration are denoted by blue
and green shaded areas, respectively. Three temperature ranges are used to categorize
the combined measurements of 2DC and CDP probes. The criteria for defining
observed cloud occurrences are described in section 2.



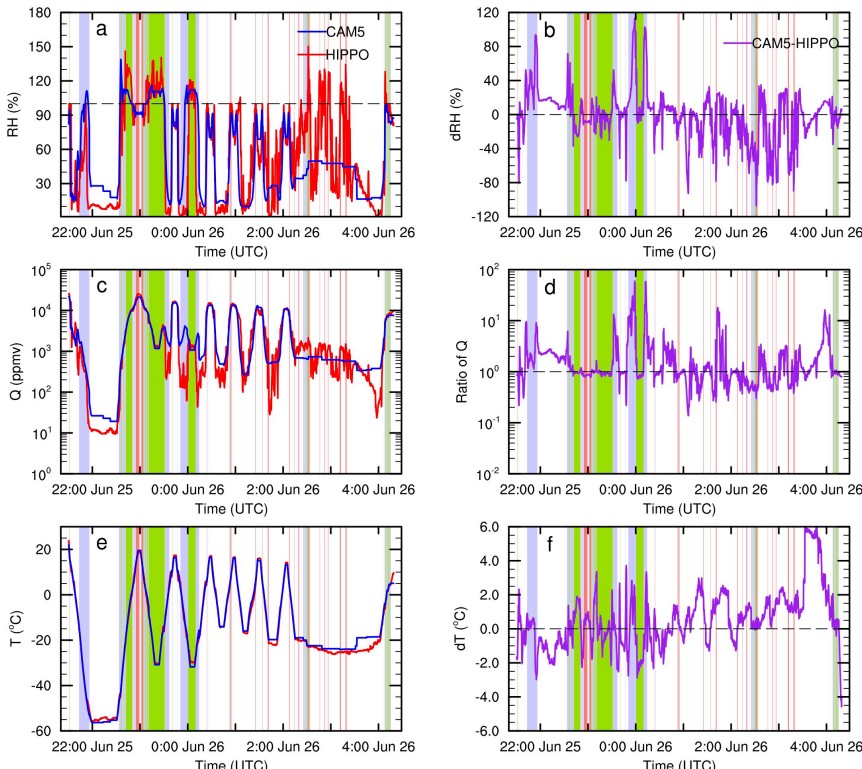


Figure 2. Spatial variabilities of RH, water vapor (Q), and temperature (T) from

CAM5 simulation and HIPPO observation (left), and their differences (right).

Absolute difference between CAM5 and HIPPO is shown for RH and T, while the

ratio between CAM5 and HIPPO is shown for Q. Model performances are denoted by

shaded vertical bars: green (red) denotes when the model captures (misses) the

observed cloud occurrences, and blue denotes when the model simulates a cloud that

is not present in the observation.





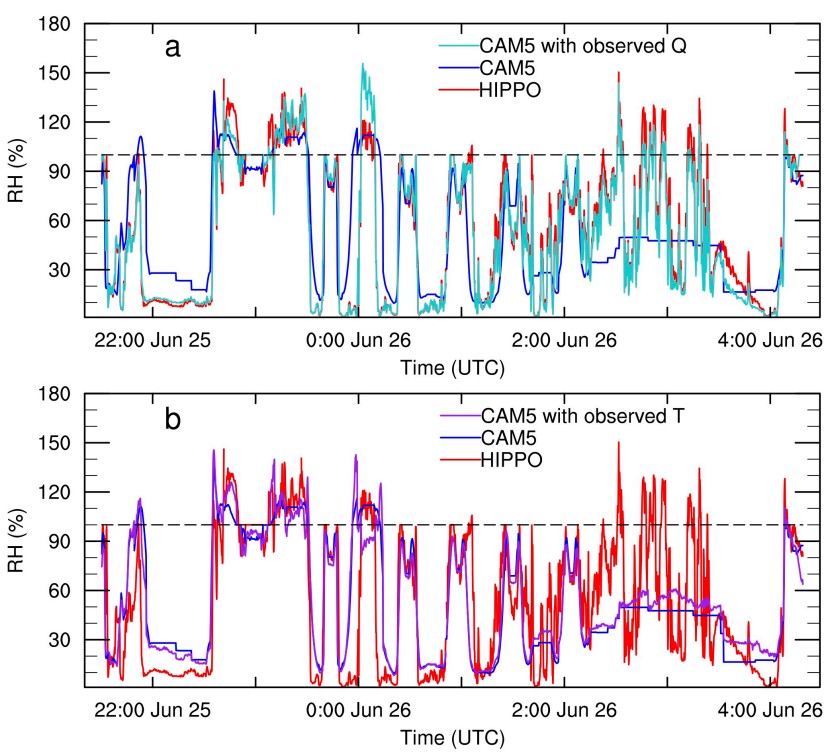


Figure 3. As Figure 2a, but for RH recalculated by replacing the model output with
either (a) observed Q or (b) observed T values.





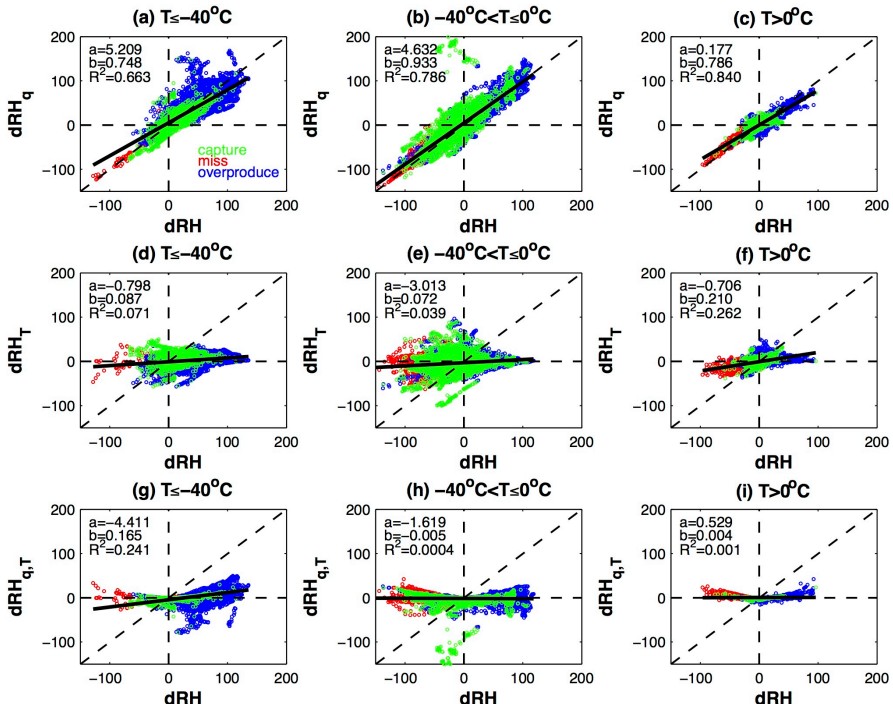

Figure 4. Corresponding (top) $d$RH$_q$ versus $d$RH, (middle) $d$RH$_T$ versus $d$RH, and (bottom) $d$RH$_{q,T}$ versus $d$RH (unit: %) for different temperature ranges. The colors indicating three types of model performances in simulating clouds as described in Fig.2: green ("captured"), red ("missed") and blue ("overproduced"). The black lines denote the linear regressions of the samples (i.e., $Y=a+b*X$), and the intercept (i.e., $a$) and slope (i.e., $b$) of the regression lines as well as the coefficient of determination (i.e., $R^2$) are shown in the legend.





1137

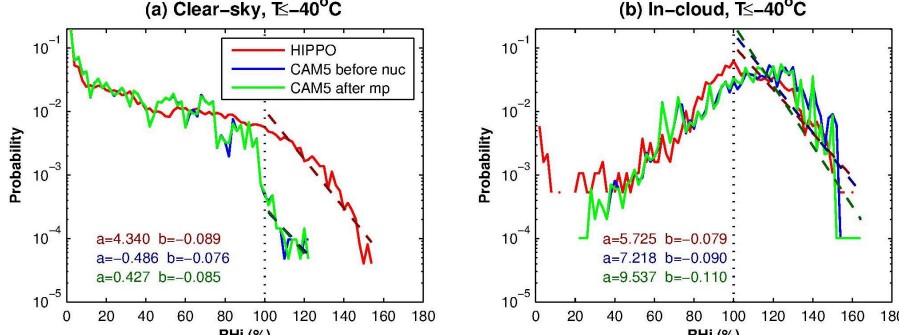

1138

Figure 5. Observed and simulated probability density functions (PDFs) of relative humidity with respect to ice (RHi, unit: %) for T≤-40°C separated into clear-sky and in-cirrus conditions. PDFs of RHi before and after cloud microphysics in the simulations are both shown. The RHi is binned by 2% for the calculation of PDF. The PDFs (when RHi>100%) follow an exponent decay: ln(PDF)=$a+b*$RHi. The values of a and b for each PDF are also shown in dark red (observed), dark blue (simulated before ice nucleaction), and dark green (simulated after cloud microphysics), respectively. Note blue lines are mostly invisible as overlaid by green lines.








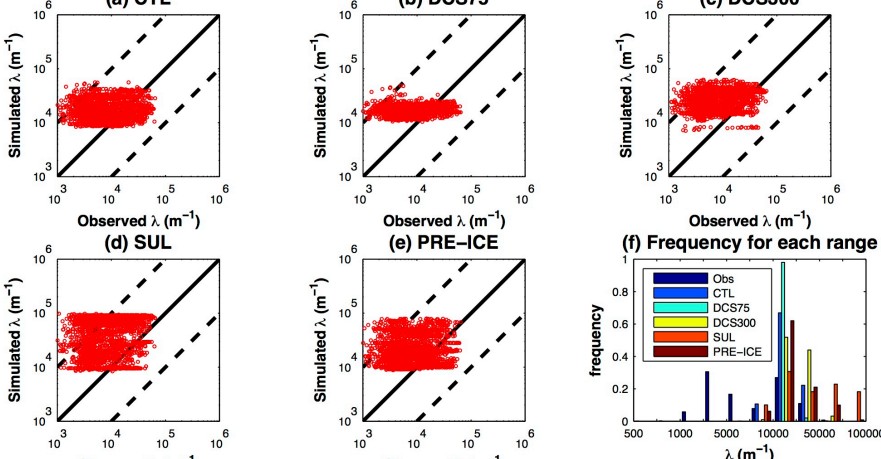


Figure 6. (a-e) Scatterplot of observed versus simulated slope parameter (λ) of the
gamma size distribution function for each experiments, and (f) the frequency of λ for
each range. Note that all the comparisons are restricted to the cases when the model
captures observed ice clouds (T≤-40 °C).






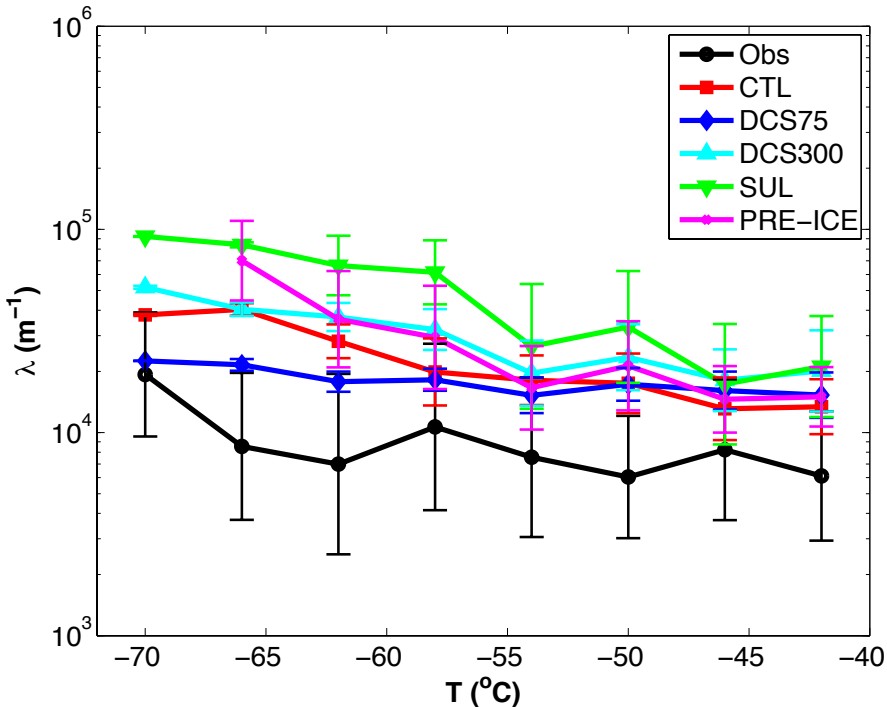


Figure 7. λ versus temperature from the measurements and simulations. The lines are

the geometric mean binned by 4°C, with the vertical bars denoting the geometric

standard deviation. Note that the comparisons are restricted to the cases when the

model captures the observed ice clouds (T≤-40 °C).











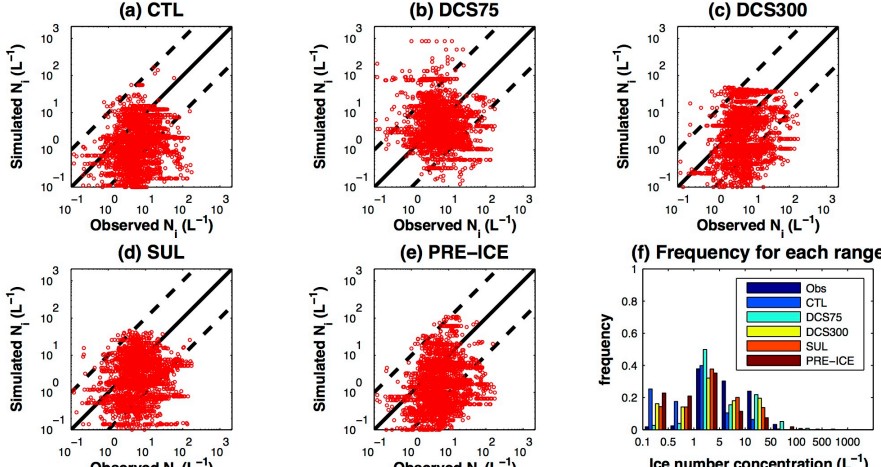


Figure 8. As Figure 6, but for the number concentrations ($N_i$) of ice crystals with
diameters larger than 75 μm for all the experiments. Note that both the comparisons
are restricted to the cases when the model captures observed ice clouds (T≤-40 °C).






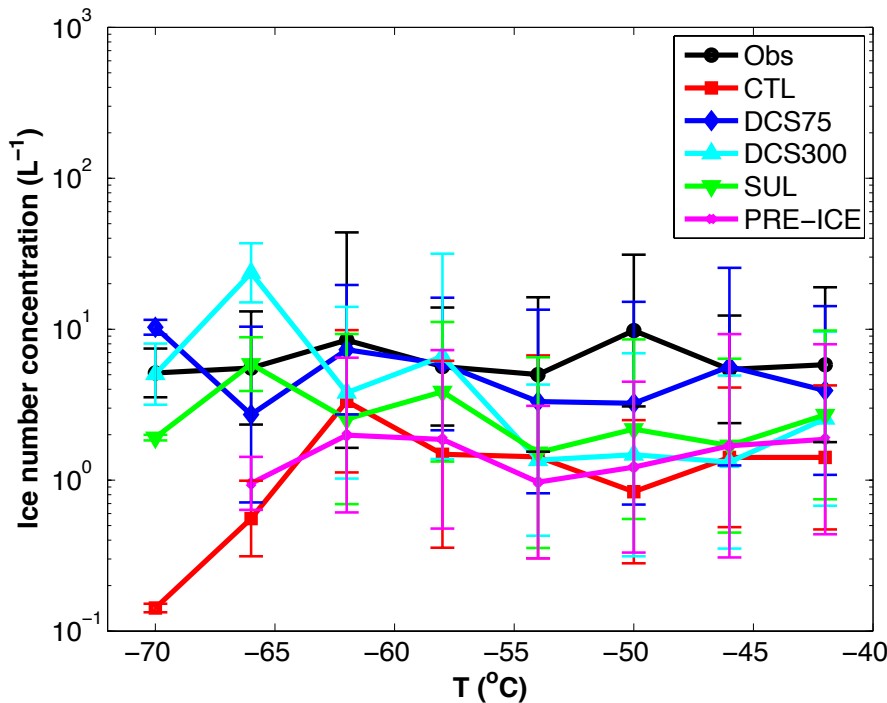


Figure 9. As Figure 7, but for $N_i$.






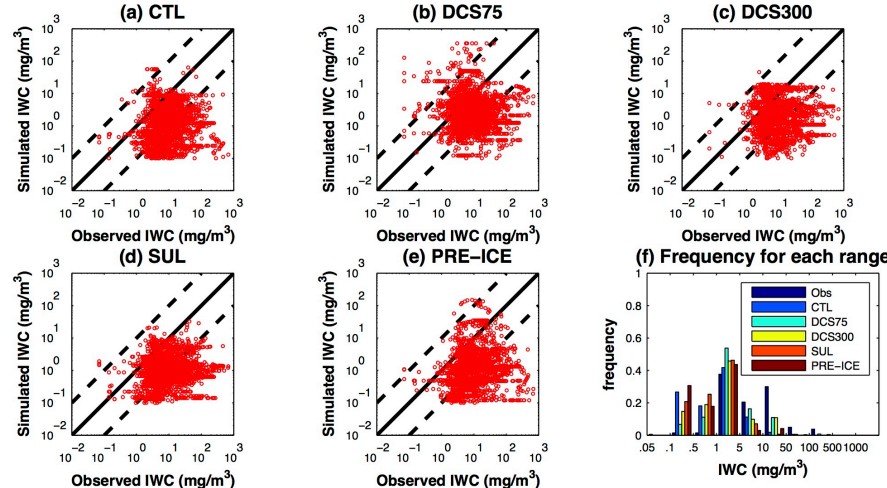


Figure 10. As Figure 8, but for the comparison of ice water content (IWC).






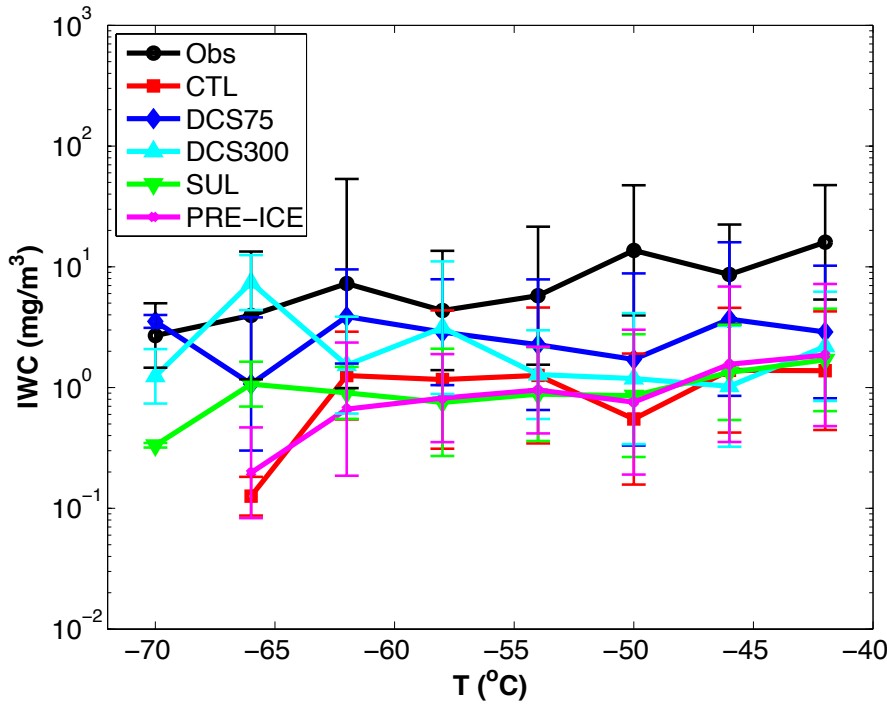


Figure 11. As Figure 9, but for ice water content (IWC) versus temperature.