# Peer review of "simulated by the Community Atmosphere Model version with"

_Atmospheric Chemistry and Physics, 2016_

## Referee Comment (RC1) · Anonymous Referee #2 · 13 Feb 2017

The Community Atmosphere Model Version 5 (CAM5) is evaluated using HIPPO measurements in this study. It shows that CAM5 can reproduce most of the observed cloud systems. This study also pointed out that the missing cloud occurrences in the model simulations are mostly attributed to the discrepancies in water vapor, and further improvements to RH variability are needed in the model.

The manuscript is overall well-written and delivers the necessary information concisely. Some revisions are needed to address the following questions before the acceptance of this manuscript:

1. Lines 269-271: Please check grammar.

2. Lines 271-272: This study uses horizontal resolution of 1.9 degree x 2.5 degree. CAM5 can be run at much higher resolution, such as 0.23 degree x 0.31 degree, which may be more appropriate for comparison with HIPPO aircraft observations and help address the over-sample issue that the authors also mentioned (lines 293-311). Please justify why a higher resolution is not used in this study.

3. Figure 1: Different colors for HIPPO observations (especially for ice clouds and warm clouds) should be used to distinguish the modeled results and observations.

4. Lines 525-526: Why there are more large ice particles at higher temperature? Is it because that it is more likely for heterogeneous nucleation (formation of larger ice crystals) to occur at higher temperature that homogeneous nucleation (formation of smaller ice crystals)?

5. Line 581-586: When all sulfate aerosol particles are available for homogeneous nucleation, it seems to me that more ice crystals with smaller size should be formed, and Ni (number of particles larger than 75 um) should decrease.

6. Lines 649-652: In previous section (Section 4.1), it is shown that the missing cloud occurrences in the model simulations are primarily ascribed to the fact that the model cannot account for the high spatial variability of observed relative humidity (RH), and that model RH biases are mostly attributed to the discrepancies in water vapor. Here it shows that when nudging both T and Q together with U and V, the model performance is even worse in terms of cloud simulations. Since the model produces clouds based on RH values, is it possible that the worse simulation of clouds in the NUG_UVTQ experiment is related to the RH threshold values used in the model?

---

## Referee Comment (RC2) · Anonymous Referee #1 · 14 Feb 2017

In this paper, the authors evaluate cloud properties as simulated with the Community Atmosphere Model Version 5 (CAM5) against observations for the HIAPER Pole-to-Pole Observations (HIPPO). To conduct a direct comparison, the model was nudged to be more representative in respect to the reanalysis. The authors show that underestimation of water vapor is responsible for most of cloud occurrence biases. They also discuss the sensitivity of autoconversion of ice to snow and ice nucleation to the modeled cloud microphysical properties as compared against observations. This paper is well written and of scientific relevance. I have a few minor comments/suggestions I would like to be addressed before publication.

[Figure]

Introduction: Page 3, line 51. I would start the introduction with: "Cirrus clouds, located at high altitudes and composed of ice crystals, are one of the key components in the climate system. They cover about 30%......"

Page 4, line 75: I suggest replace "higher" with "high" (since there are no mention yet what nucleates at lower supersaturations), and the give a typical range of supersaturations.

Page 4, line 83: Replace "ice microphysics" with "ice microphysical processes"

Page 5, line 110: What is meant by fast measurements? High frequency measurements?

Page 67, line 140. What about observations of water vapor? Since much of the analysis is in regard to the relative humidity and supersaturation, I think the observations of water vapor should be included as well.

Page 12, line 248: Replace "the" with "for"

Page 12, line 256, add "a" between "includes version", so that "includes a version"

Page 16, line 340. "Reword CAM5 is able to better simulate cloud systems . . . ."

Page 21, line 465. I suggest to rewrite: "The model is capable to simulate the occurrences of ice . . . . . ." i.e. remove the reference to comparison with observations, since the model does a poor job in simulating supersaturation in clear sky.

Page 568, line 568 (or figure 8F). The point of DCS75 and PRE-ICE can produce Ni>50 L-1 is hard to see because the figure is too small.

Page 31, line 678: Replace "which nudge the" with "with nudged". Page 32, line 688. Remove "and" before 86.1% Page 32, line 691: Remove "of" Page 32, line 705: Add "to" so that "The model is mostly able to reproduce..." Page 34, line 735. Suggest adding "global" such that ". . ..future global model. . .."

Page 34, line 746. A recently published paper by Eidhammer et al. (2017) describes the implementation of the single ice category in CAM5. I suggest including this citation on line 746.

Eidhammer, T., H. Morrison, D. Mitchell, A. Gettelman, and E. Erfani, 2017: Improvements in Global Climate Model Microphysics Using a Consistent Representation of Ice Particle Properties. J. Climate, 30, 609–629, doi: 10.1175/JCLI-D-16-0050.1.
* * *

---

## Author Comment (AC1) · 10 Mar 2017

We thank the two anonymous reviewers for their valuable comments and constructive suggestions on the manuscript. Below, we explain how the comments and suggestions are addressed and make note of the revision in the revised manuscript.

**Reviewer #2**

*The Community Atmosphere Model Version 5 (CAM5) is evaluated using HIPPO measurements in this study. It shows that CAM5 can reproduce most of the observed cloud systems. This study also pointed out that the missing cloud occurrences in the model simulations are mostly attributed to the discrepancies in water vapor, and further improvements to RH variability are needed in the model.*
*The manuscript is overall well-written and delivers the necessary information concisely. Some revisions are needed to address the following questions before the acceptance of this manuscript:*

Reply: We thank the reviewer for his/her helpful comments. The text and figures are revised as the reviewer suggested.

*1. Lines 269-271: Please check grammar.*
Reply: We changed the sentence to "We also conduct two experiments, one with only U and V nudged (referred to as NUG_UV) and the other with U, V, T and water vapor (Q) nudged (referred to as NUG_UVTQ)" in the revised manuscript.

*2. Lines 271-272: This study uses horizontal resolution of 1.9 degree x 2.5 degree. CAM5 can be run at much higher resolution, such as 0.23 degree x 0.31 degree, which may be more appropriate for comparison with HIPPO aircraft observations and help address the over-sample issue that the authors also mentioned (lines 293-311). Please justify why a higher resolution is not used in this study.*

Reply: Although CAM5 can be run at higher resolutions, we choose the resolution of 1.9 degree x 2.5 degree in this study because this resolution is still widely used in CAM5 simulations and in other climate model simulations. We prefer to first evaluate the model performance at this resolution before we move to higher resolutions. As mentioned in Section 6 (Discussion and Conclusions), understanding the resolution dependence of model results is also desirable and we plan to investigate it in the near future. In particular, as the reviewer pointed out, using higher resolutions will help address the

over-sample issue in the comparison with observations in the present study, which will also be examined. We have added the above justifications in the revised manuscript.

*3. Figure 1: Different colors for HIPPO observations (especially for ice clouds and warm clouds) should be used to distinguish the modeled results and observations.*

Reply: Following the reviewer's suggestion, different colors for HIPPO observations (i.e., violet and brown for ice clouds and warm clouds, respectively) are used to distinguish the modeled results and observations in the revised manuscript.

*4. Lines 525-526: Why there are more large ice particles at higher temperature? Is it because that it is more likely for heterogeneous nucleation (formation of larger ice crystals) to occur at higher temperature that homogeneous nucleation (formation of smaller ice crystals)?*

Reply: We thank the reviewer for the comment. The relationship between slope parameter and temperature depends not only on ice nucleation but also on ice crystal growth. With increasing temperature (or decreasing height), there is more water vapor available for the growth of ice crystals (due to increased saturation vapor pressure), which can partly explain the decreasing trend of slope parameter with temperature. In addition, as mentioned by the reviewer, it is more likely for heterogeneous ice nucleation to occur at higher temperature than homogeneous nucleation and the former process tends to form less ice crystals to produce larger ice crystals (due to less competition for the available water vapor). In the revised manuscript, we revised the explanation to make it clearer: "Such a feature is mainly due to more small ice particles at lower temperatures, which can be explained by less water vapor available for ice crystal growth as well as more ice crystals formed from nucleation at lower temperatures (more likely from homogeneous nucleation than from heterogeneous nucleation) (Eidhammer et al., 2014)."

*5. Line 581-586: When all sulfate aerosol particles are available for homogeneous nucleation, it seems to me that more ice crystals with smaller size should be formed, and Ni (number of particles larger than 75 um) should decrease.*

Reply: We thank the reviewer for pointing out the consistency between ice crystal size (in term of slope parameter) and $N_i$ (number concentration of ice crystals larger than 75 µm). When all sulfate aerosol particles are available for homogeneous nucleation, the slope parameter for the gamma size distribution is much larger in SUL, indicating a larger

fraction of ice crystals with smaller size and a smaller fraction of ice crystals with larger size (Figure 7). However, as total ice crystal number concentration in SUL is much higher (one to two orders magnitude larger) than that in CTL, especially at lower temperature, overall $N_i$ in SUL does not decrease but increases compared to that in CTL (Figure 9). In the revised version, we added the explanation for the difference of $N_i$ between SUL and CTL: "With the removal of the lower size limit (0.1 μm diameter) of sulfate aerosol particles for homogeneous nucleation in the experiment SUL, simulated $N_i$ is significantly higher than that in CTL because of the substantial increase in the total ice crystal number concentration in SUL, although the slope parameter in SUL is larger indicating a smaller fraction of ice crystals with larger sizes (e.g., larger than 75 μm)."

***6. Lines 649-652: In previous section (Section 4.1), it is shown that the missing cloud occurrences in the model simulations are primarily ascribed to the fact that the model cannot account for the high spatial variability of observed relative humidity (RH), and that model RH biases are mostly attributed to the discrepancies in water vapor. Here it shows that when nudging both T and Q together with U and V, the model performance is even worse in terms of cloud simulations. Since the model produces clouds based on RH values, is it possible that the worse simulation of clouds in the NUG_UVTQ experiment is related to the RH threshold values used in the model?***

Reply: We thank the review for this constructive comment. Indeed, the model simulation of cloud occurrences is sensitive to RH threshold used in the calculation of cloud fraction. For instance, with a smaller RH threshold ($RH_{min}$), the model can simulate larger cloud fraction and produce cloud occurrences at lower grid-mean RH. In NUG_UVTQ, although Q is nudged in the model, the model simulates worse cloud occurrences. It is possible that discrepancy in the cloud fraction scheme (e.g., the chosen $RH_{min}$) may also partly contribute to the degradation of simulation. Following the reviewer's comment, we added the discussion in Section 5 in the revised manuscript: "The bias in cloud occurrences may also be related to the RH threshold values used in the cloud fraction scheme in the model (Park et al., 2014), and further study is needed to address the model sensitivity to the RH threshold values."

---

## Author Comment (AC2) · 10 Mar 2017

We thank the two anonymous reviewers for their valuable comments and constructive suggestions on the manuscript. Below, we explain how the comments and suggestions are addressed and make note of the revision in the revised manuscript.

**Reviewer #1**

*In this paper, the authors evaluate cloud properties as simulated with the Community Atmosphere Model Version 5 (CAM5) against observations for the HIAPER Pole-to-Pole Observations (HIPPO). To conduct a direct comparison, the model was nudged to be more representative in respect to the reanalysis. The authors show that underestimation of water vapor is responsible for most of cloud occurrence biases. They also discuss the sensitivity of autoconversion of ice to snow and ice nucleation to the modeled cloud microphysical properties as compared against observations. This paper is well written and of scientific relevance. I have a few minor comments/suggestions I would like to be addressed before publication.*

Reply: We thank the reviewer for constructive review and encouraging comments. The text and figures are revised as the reviewer suggested.

*Introduction: Page 3, line 51. I would start the introduction with: "Cirrus clouds, located at high altitudes and composed of ice crystals, are one of the key components in the climate system. They cover about 30%......"*
Reply: Done.

*Page 4, line 75: I suggest replace "higher" with "high" (since there are no mention yet what nucleates at lower supersaturations), and the give a typical range of supersaturations.*

Reply: We changed the text "Homogeneous nucleation generally requires higher supersaturation" to "Homogeneous nucleation generally requires high ice supersaturation typically of 40%-60%" in the revised manuscript.

*Page 4, line 83: Replace "ice microphysics" with "ice microphysical processes"*
Reply: Done.

*Page 5, line 110: What is meant by fast measurements? High frequency measurements?*

Reply: Yes, we meant high frequency measurements. We changed "fast measurements" to "high frequency measurements" in the revised manuscript.

*Page 67, line 140. What about observations of water vapor? Since much of the analysis is in regard to the relative humidity and supersaturation, I think the observations of water vapor should be included as well.*

Reply: Thank the reviewer for this comment. We included "water vapor" in describing "measurements of ambient environmental conditions" in the revised manuscript.

*Page 12, line 248: Replace "the" with "for"*
Reply: Done.

*Page 12, line 256, add "a" between "includes version", so that "includes a version"*
Reply: Done.

*Page 16, line 340. "Reword CAM5 is able to better simulate cloud systems ...."*
Reply: Done.

*Page 21, line 465. I suggest to rewrite: "The model is capable to simulate the occurrences of ice ……" i.e. remove the reference to comparison with observations, since the model does a poor job in simulating supersaturation in clear sky.*
Reply: Done.

*Page 568, line 568 (or figure 8F). The point of DCS75 and PRE-ICE can produce Ni>50 L-1 is hard to see because the figure is too small.*

Reply: Thank you for pointing out this. We added an inset with rescaled axes in Figure 8f to illustrate the frequency of $N_i$ when $N_i$ >50 L$^{-1}$. From the inset, it is clear that DCS75 and PRE-ICE can produce $N_i$ >50 L$^{-1}$.

*Page 31, line 678: Replace "which nudge the" with "with nudged". Page 32, line 688. Remove "and" before 86.1% Page 32, line 691: Remove "of" Page 32, line 705: Add "to" so that "The model is mostly able to reproduce…" Page 34, line 735. Suggest adding "global" such that "….future global model…."*

Reply: All the suggested revisions are done in the revised manuscript.

***Page 34, line 746. A recently published paper by Eidhammer et al. (2017) describes the implementation of the single ice category in CAM5. I suggest including this citation on line 746.***
***Eidhammer, T., H. Morrison, D. Mitchell, A. Gettelman, and E. Erfani, 2017: Improvements in Global Climate Model Microphysics Using a Consistent Representation of Ice Particle Properties. J. Climate, 30, 609–629, doi: 10.1175/JCLI-D-16-0050.1.***

Reply: We thank the reviewer for pointing us to the work of Eidhammer et al. (2017), which is very relevant to our study. We have cited their study for references in the revised version.